

# Climatic variation in Africa and Europe has combined effects on timing of spring migration in a long-distance migrant Willow Warbler *Phylloscopus trochilus*

Magdalena Remisiewicz[1,2] and Les G. Underhill[2,3]

[1] Bird Migration Research Station, Faculty of Biology, University of Gdańsk, Gdańsk, Poland
[2] Animal Demography Unit, Department of Biological Sciences, University of Cape Town, Cape Town, South Africa
[3] Biodiversity and Development Institute, Cape Town, South Africa

Corresponding author
Magdalena Remisiewicz,
magdalena.remisiewicz@biol.ug.edu.pl

## ABSTRACT

**Background**. The arrival of many species of migrant passerine in the European spring has shifted earlier over recent decades, attributed to climate change and rising temperatures in Europe and west Africa. Few studies have shown the effects of climate change in both hemispheres though many long-distance migrants use wintering grounds which span Africa. The migrants' arrival in Europe thus potentially reflects a combination of the conditions they experience across Africa. We examine if the timing of spring migration of a long-distance migrant, the Willow Warbler, is related to large-scale climate indices across Africa and Europe.

**Methods**. Using data from daily mistnetting from 1 April to 15 May in 1982–2017 at Bukowo (Poland, Baltic Sea coast), we developed an Annual Anomaly metric (AA, in days) to estimate how early or late Willow Warblers arrive each spring in relation to their multi-year average pattern. The Willow Warblers' spring passage advanced by 5.4 days over the 36 years. We modelled AA using 14 potential explanatory variables in multiple regression models. The variables were the calendar year and 13 large-scale indices of climate in Africa and Europe averaged over biologically meaningful periods of two to four months during the year before spring migration.

**Results**. The best model explained 59% of the variation in AA with seven variables: Northern Atlantic Oscillation (two periods), Indian Ocean Dipole, Southern Oscillation Index, Sahel Precipitation Anomaly, Scandinavian Index and local mean temperatures. The study also confirmed that a long-term trend for Willow Warblers to arrive earlier in spring continued up to 2017.

**Discussion**. Our results suggest that the timing of Willow Warbler spring migration at the Baltic Sea coast is related to a summation of the ecological conditions they had encountered over the previous year during breeding, migration south, wintering in Africa and migration north. We suggest these large-scale climate indices reflect ecological drivers for phenological changes in species with complex migration patterns and discuss the ways in which each of the seven climate indices could be related to spring migration at the Baltic Sea coast.

## INTRODUCTION

Environmental conditions have changed in recent decades as a consequence of global climate change and these changes have had knock-on effects in the timing of the life stages of plants and animals (e.g., *Walther et al., 2002*; *Vose, Easterling & Gleason, 2005*; *Peñuelas, 2009*). Phenological changes in bird migrations are one of the best-described animal responses to climate change in the northern hemisphere (e.g., *Miller-Rushing et al., 2008*; *Wood & Kellermann, 2015*; *Miles et al., 2017*; *Zaifman et al., 2017*; *Lehikoinen et al., 2019*). The timing of many passerines' arrival in the European spring has shifted earlier in recent decades (e.g., *Sokolov et al., 1998*; *Hüppop & Hüppop, 2003*; *Vähätalo et al., 2004*; *Tøttrup, Thorup & Rahbek, 2006a*; *Miles et al., 2017*), which is mostly linked to increasing spring temperatures (*Lehikoinen et al., 2019*).

Most long-distance migrants breeding in Europe winter in the southern hemisphere, but few studies have sought to understand the effects on migrants of climate change in both hemispheres (*Altwegg et al., 2012*; *Ockendon, Leech & Pearce-Higgins, 2013*; *Bussière, Underhill & Altwegg, 2015*). Advances in spring arrival in Europe have been related to the winter and annual North Atlantic Oscillation Index (NAOI), which reflect weather patterns in western Europe and northern Africa (*Forchhammer, Post & Stenseth, 2002*; *Møller, 2002*; *Cotton, 2003*; *Stenseth et al., 2003*; *Ahola et al., 2004*; *Gordo, Barriocanal & Robson, 2011*; *Grimm et al., 2015*). NAOI explained only 0%–6% of the variance in the phenology of spring migration in 23 species of migrants in Helgoland, Germany (*Haest, Hüppop & Bairlein, 2018*), probably because this index does not reflect the conditions the migrants experience farther south. Most studies have used temperatures and rainfall from locations the birds likely visit (*Saino & Ambrosini, 2008*; *Pasinelli et al., 2011*; *Halupka & Halupka, 2017*) and the Normalized Difference Vegetation Index (NDVI) (*Tøttrup et al., 2012*; *Jørgensen et al., 2016*; *Thorup et al., 2017*) as proxies for the ecological conditions that the migrants encounter in Africa. Few studies have used the Sahel Precipitation Anomaly (SAH) (e.g., *Zwarts et al., 2009*; *Tobolka et al., 2018*) as a proxy for rainfall in the Sahel, where many European migrants stop over or stay for winter. The relationship between the timing of migration and the El-Niño-Southern Oscillation (ENSO/SOI) has been marginally explored (e.g., *Stenseth et al., 2003*; *MacMynowski & Root, 2007*; *Miller-Rushing et al., 2008*), as has the Indian Ocean Dipole (IOD) (e.g., *Hušek et al., 2008*; *Zduniak et al., 2010*; *Tryjanowski, Stenseth & Matysioková, 2013*; *Tobolka et al., 2018*). The SOI and IOD are continental-scale weather anomalies which are correlated with rainfall and temperatures in the southern hemisphere (*Black, 2005*; *Marchant et al., 2006*), thus they might serve as useful indices that reflect the conditions which migrant European birds experience in eastern and southern Africa.

Changes in the timing of a bird's life stages which occur on the southern non-breeding grounds will likely have carry-over effects on subsequent stages, including stages on the northern breeding grounds (*Peach, Baillie & Underhill, 1991*; *Gordo, 2007*; *Barshep et al., 2011*; *Conklin & Battley, 2012*; *Tobolka et al., 2018*; *Tomotani et al., 2018*). Birds which migrate between Eurasia and Africa adjust the timing of their moult and pre-migratory fattening to the conditions at their non-breeding grounds in Africa (*Salewski et al., 2004*;

*Zwarts et al., 2009*; *Altwegg et al., 2012*; *Tomotani et al., 2018*; *Remisiewicz et al., 2019*). Weather, which determines food availability on the non-breeding grounds, affects pre-migratory preparations by the migrants, and thus might influence their departure time and migration strategy (*Katti & Price, 1999*; *Saino & Ambrosini, 2008*; *Studds & Marra, 2011*; *Ouwehand & Both, 2017*). Environmental conditions at autumn and spring migration stopover sites also influence the timing of spring arrival at the breeding grounds (*Tøttrup et al., 2012*; *Halupka et al., 2017*). Demographic parameters at the breeding sites of long-distance migrants, such as White Storks *Ciconia alba* and Red-backed Shrikes *Lanius collurio*, reflect a carry-over effect of environmental conditions at their stopover sites and at their wintering grounds in Africa (*Hušek et al., 2008*; *Tobolka et al., 2018*).

Certain long-distance migrant passerines, such as Red-backed Shrike, Barn Swallow *Hirundo rustica*, European Reed Warbler *Acrocephalus scirpaceus*, Sedge Warbler *A. schoenobaenus*, Spotted Flycatcher *Muscicapa striata*, Garden Warbler *Sylvia borin* and Willow Warbler *Phylloscopus trochilus*, breed in Europe and occupy non-breeding grounds which span most of Africa south of the Sahara Desert (*Cepák et al., 2008*; *Fransson & Hall-Karlsson, 2008*; *Bairlein et al., 2014*; *Valkama et al., 2014*). In many migrants, the populations or subspecies which use different wintering grounds meet at the breeding grounds in Europe, such as White Stork (*Tobolka et al., 2018*), Blackcap *Sylvia atricapilla* and Willow Warbler (*Bensch et al., 2009*; *Bensch, Liedvogel & Åkesson, 2011*). Thus, these species' patterns of spring arrival in Europe should reflect a combination of the diverse conditions they experience across their non-breeding areas. So far, few studies have examined if and how the combined effects of conditions which different migratory populations of a species encounter at their various non-breeding grounds and stopover sites affect their arrival and performance at the breeding grounds (*Saino et al., 2007*; *Tobolka et al., 2018*). Uncovering the complex relationships between climatic variation at a global scale and the migration patterns of migratory birds is crucial to understand the effects of climate change on the seasonal timing and population trends of these wide-ranging species (*Hayhow et al., 2014*). This study explores this phenological pattern of a long-distance migrant passerine, the Willow Warbler, which we chose as a case study. We aimed to examine if the timing of the Willow Warbler's spring migration at the Baltic Sea coast had responded to large-scale climate variables affecting the widespread non-breeding grounds in the year before their migration.

## MATERIALS & METHODS

### Study species

Our choice of the Willow Warbler as a study species was motivated by the availability of detailed information about the timing of its spring migration to northern Europe at a stopover site on the Baltic Sea coast. Different populations of this species migrate to their breeding grounds in Scandinavia and the Baltic region 5,000–12,000 km from their wintering grounds south of the Sahara Desert in west, central, east and southeast Africa (*Cramp & Brooks, 1992*; *Fransson & Hall-Karlsson, 2008*; *Zwarts et al., 2009*; *Valkama et al., 2014*; *Lerche-Jørgensen et al., 2017*; *Maciąg et al., 2017*).

Two subspecies of Willow Warbler breed in northern and central Europe, then migrate across the Baltic Sea region, including Bukowo. *Phylloscopus trochilus trochilus* breeds in southwestern Scandinavia, most of Poland, and western and southern Europe, then migrates mostly to west and central sub-Saharan Africa; *Ph. t. acredula* breeds in northeastern Scandinavia, eastern Poland and northeast of these areas, then migrates mostly to central, eastern and southern Africa (Fig. 1) (*Tomiałojć & Stawarczyk, 2003*; *Bensch, Bengtsson & Åkesson, 2006*; *Fransson & Hall-Karlsson, 2008*; *Bensch et al., 2009*; *Zwarts et al., 2009*; *Valkama et al., 2014*; *Lerche-Jørgensen et al., 2017*). The proportions of the two Willow Warbler subspecies which migrate through Bukowo are difficult to estimate because the differences in colour and size between them are clinal (*Bensch et al., 2009*). Willow Warblers which stop over at the Polish coast originate from Sweden and Finland, and probably from other Baltic countries (*Maci̦ag et al., 2017*). Given that Bukowo lies close to the migratory divide between the two subspecies of Willow Warblers across Sweden and eastern Poland (Fig. 1) (*Bensch et al., 2009*), we assumed that we caught both subspecies. Willow Warblers from the western part of the breeding grounds migrate mostly to west Africa, but the proportion of Willow Warblers which use non-breeding grounds in eastern and southeastern Africa increases eastwards across their breeding grounds (*Zwarts et al., 2009*). The farthest ringing recoveries of Willow Warblers ringed at the Polish coast and elsewhere in Poland come from the Iberian Peninsula in the west and Egypt in the east, thus these recoveries reflect the species' migration routes, not their wintering grounds south of the Sahara (*Maci̦ag et al., 2017*). Willow Warblers from Sweden and Finland, which pass through Bukowo, use wintering grounds in west, central, east and southeast Africa, as far as South Africa (*Fransson & Hall-Karlsson, 2008*; *Valkama et al., 2014*). Based on these established links, the Willow Warblers migrating along the Polish coast in spring have non-breeding grounds in western, eastern and southern Africa. We therefore expected that the timing of Willow Warblers' northward migration at Bukowo would be affected by the large-scale climate indices which reflect conditions across Africa.

Willow Warblers inhabit trees and bushes, from wooded savannah to riparian thickets, and feed mainly on insects and spiders which they pick from vegetation, supplemented by berries (*Cramp & Brooks, 1992*; *Zwarts et al., 2009*). They arrive in central and northern Europe, including the Baltic Sea coast, in April–May, breed in June–July, depart from the breeding grounds in August–September, migrate south in August–October and occupy their non-breeding grounds in Africa in November–March (Fig. 1) (*Cramp & Brooks, 1992*; *Herremans, 1999*; *Tomiałojć & Stawarczyk, 2003*; *Dean, 2005*; *Zwarts et al., 2009*). Our study site is in the northern hemisphere, and we use boreal seasons throughout this paper: "spring migration" is April–May, "autumn migration" is August–October and "winter" is November–March.

## Study site and sampling

We analysed the daily numbers of Willow Warblers ringed during spring migration using standardised mistnetting protocols at Bukowo ringing station on the Baltic Sea coast (54°20′13″N, 16°14′36″E) in 1982–2017 (Table S1, Supplementary Information). Migrating passerines were caught in mist nets located on spits between the Baltic Sea and

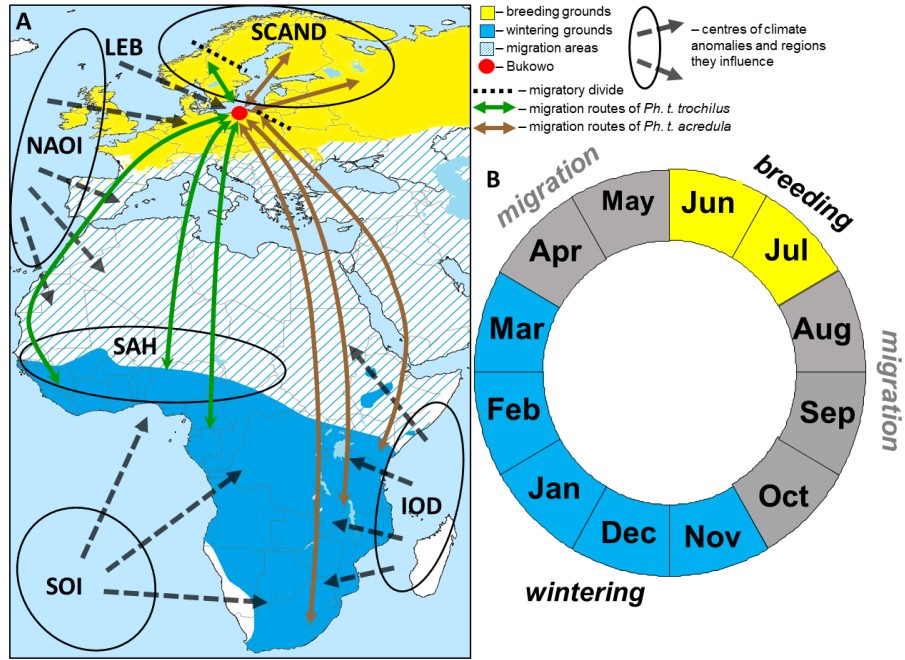

**Figure 1 The hypothetical pattern of the influences of the climate variables we used in the study in relation to the Willow Warbler range (A) and life cycle (B).** (A) Approximate migration routes and migratory divides (after *Bensch et al., 2009*) for the two subspecies that pass through Bukowo, and centres of the large-scale climate anomalies and regions they influence. Abbreviations as in Table 1. (B) Four main periods in an annual cycle of Willow Warbler. This figure ("Willow Warbler and climate indices") is a derivative of "*Phylloscopus trochilus* Range Map.png" by Keith W. Larson, licenced under CC-BY-SA-3.0, and is licenced under CC-BY-SA-3.0 by Magdalena Remisiewicz.

the Bukowo and Kopań coastal lakes. The habitat composes mixed coastal and riparian forests with an undergrowth of fruit bushes, and willow bushes and reedbeds along the margins of the lakes (*Nowakowski et al., 2012*). Birds were caught from dawn until dusk using 35 to 59 8-m long mist nets. The number of nets was stable during each season, but changed between years. Mistnetting and ringing were conducted daily from dawn to dusk, from the last week of March to mid-May. All birds caught were ringed, aged and measured, then released. The same standard fieldwork protocol was used annually (*Busse & Meissner, 2015*). In spring, adult and immature Willow Warblers are all in the same plumage and were aged as "full grown" (*Svensson, 1992*; *Demongin, 2016*), thus we treated them jointly in our study. Our analysis is based on birds ringed between 1 April and 15 May. The earliest date a Willow Warbler has been captured at Bukowo was 2 April 2017. We took 15 May as the end of the study period; after this date, frequent recaptures of the same birds indicated that they were local breeders (Supplement 2). In our analyses we used the daily totals of ringed Willow Warblers based on the first captures of Willow Warblers and ignoring recaptures. This protocol enabled us to quantify the timing of Willow Warblers' spring migration at the Baltic Sea coast of Poland. Our catching and ringing of birds was conducted with the authorisation of the Polish Academy of Sciences, with the approval of the General

Directorate for Environmental Protection, Poland (DZP-WG.6401.03.97.2017.jro). Field research at Bukowo was approved by the Marine Office, Słupsk (OW-A-510/87/17/ds).

## Climate indices

We used 13 large-scale and local climate indices as proxies for the ecological conditions (*Stenseth et al., 2003*) which the Willow Warblers had encountered during the year preceding their spring migration at the Baltic Sea coast (Fig. 1, Table 1). These indices reflect conditions at the breeding grounds in north Europe, on southward migration, at the wintering grounds in Africa and on northward migration from Africa across Europe. The Scandinavian pattern (SCAND) reflects the primary circulation centre over Scandinavia; its positive phase in summer is associated with low precipitation and high temperatures (*Bueh & Nakamura, 2007*). The North Atlantic Oscillation Index (NAOI) reflects the weather patterns over Europe and west Africa; positive winter values are related to warm and wet winters in west Africa and early springs in western Europe (*Hurrell, 1995*). The Sahel Precipitation Anomaly (SAH) reflects rainfall in the Sahel region; positive values are associated with higher-than-average rainfall (*Munemoto & Tachibana, 2012*). The El Niño-Southern Oscillation Index (SOI) reflects the El Niño and La Niña phases of circulation in the eastern Pacific Ocean, which affect weather in southern Africa; negative SOI values indicate an El Niño, associated with below-average rainfall across southern Africa (*Saji & Vinayachandran, 1999*; *Stenseth et al., 2003*). The Indian Ocean Dipole (IOD) is the gradient between sea surface temperatures in the eastern and western Indian Ocean; when the IOD is positive east and southeastern Africa receive above-average rainfall (*Marchant et al., 2006*; *Schott, Xie & McCreary, 2009*). The IOD is sometimes in phase with SOI but is frequently independent (*Ashok, Guan & Yamagata, 2003*; *Marchant et al., 2006*; *Schott, Xie & McCreary, 2009*). We downloaded monthly values of these indices for 1981–2017 from databases of the US National Oceanic and Atmospheric Administration, National Weather Service, Climate Prediction Center (http://www.cpc.ncep.noaa.gov/). We also downloaded mean daily temperatures for Łeba (54°45′N, 17°32′E), the nearest coastal weather station to Bukowo, located 95 km east and similarly exposed to local marine weather, from the European Climate Assessment and Dataset (http://www.ecad.eu). From these daily temperatures we calculated the mean temperature at Łeba for April–May (LEBT Apr–May), which increased on average by 1.5 °C over the 36 years of our study (Tables S2 and S3). We used this temperature as a proxy for local ecological conditions on the birds' arrival in spring at Bukowo, as in other studies on Willow Warbler spring phenology (*Sokolov et al., 1998*; *Palm et al., 2009*).

The selected weather indices represented climatic variation across the wide geographical range which different populations of Willow Warblers visit during their annual cycle. We constructed an annual time series for each climate variable by selecting the biologically meaningful months (Fig. 1) and averaging the values for these months: (1) spring arrival at the Baltic Sea coast occurs in April–May, we therefore averaged NAOI and mean temperature in Łeba for April–May of the same year; (2) wintering in Africa occurs from November of the previous year to March, so we averaged NAOI, SAH, IOD and SOI for the preceding November–March; (3) migration from northern Europe to Africa occurs in

**Table 1** Fourteen explanatory variables and the response variable (AA), used in modelling of Willow Warbler spring migration (1 April–15 May) in 1982–2017 at Bukowo ringing station (N Poland).

| No | Abbreviations used in the text | Variable | $\chi^2$ | $P$ |
|----|---------|---------|------|------|
| 1 | Year | Year, as the year number from 1982 = Year 1 | – | – |
| 2 | LEB T Apr–May | Apr–May mean of the daily means of temperatures in Łeba | 12.20 | 0.27 |
| 3 | NAOI Apr–May | Apr–May mean of the Northern Atlantic Oscillation Index | 12.59 | 0.25 |
| 4 | SCAND Apr–May | Apr–May mean of the Scandinavian Pattern Index | 10.11 | 0.43 |
| 5 | NAOI Nov–Mar | Nov–Mar mean of the Northern Atlantic Oscillation Index | 3.21 | 0.98 |
| 6 | SAH Nov–Mar | Nov–Mar mean of the Sahel Precipitation Anomaly | 13.39 | 0.20 |
| 7 | IOD Nov–Mar | Nov–Mar mean of the Indian Ocean Dipole | 7.19 | 0.71 |
| 8 | SOI Nov–Mar | Nov–Mar mean of the Southern Oscillation Index | 1.33 | 0.99 |
| 9 | NAOI Aug–Oct | Aug–Oct mean of the Northern Atlantic Oscillation Index | 13.79 | 0.18 |
| 10 | SAH Aug–Oct | Aug–Oct mean of the Sahel Precipitation Anomaly | 7.68 | 0.66 |
| 11 | IOD Aug–Oct | Aug–Oct mean of the Indian Ocean Dipole | 11.33 | 0.33 |
| 12 | SOI Aug–Oct | Aug–Oct mean of the Southern Oscillation Index | 6.23 | 0.80 |
| 13 | NAOI Jun–Jul | Jun–Jul mean of the Northern Atlantic Oscillation Index | 2.65 | 0.99 |
| 14 | SCAND Jun–Jul | Jun–Jul mean of the Scandinavian Pattern Index | 14.14 | 0.17 |
| 15 | AA | Annual Anomaly (AA) in spring at Bukowo | 9.88 | 0.45 |

Notes.

$\chi^2$, $P$ – results of the Ljung-Box portmanteau for a lag up to 10 years used to test the null hypothesis of a lack of autocorrelations in each data series. The weather indices are presented in reverse order of life stages during the year preceding their April–May migration through Bukowo.

August–October of the previous year, thus we averaged SCAND, NAOI, SAH, IOD and SOI for the previous August–October; (4) the previous stay at the breeding grounds occurs in the preceding June–July, we thus averaged SCAND and NAOI for June–July of the previous year. We generated 13 climate indices (Table 1) and used these as explanatory variables in a multiple regression model to estimate the influence of climate on the timing of Willow Warblers' arrival in spring at the Baltic Sea coast. The SAH in August–October and IOD in November–March increased, and June–July NAOI decreased over the 36 years of our study (Tables S2 and S3). The averaged NAOI indices in subsequent periods were not correlated, nor were the SAH and SCAND indices (Table S2). The IOD in November–March was correlated with IOD in August–October, and with NAOI and SAH in this period, and a few indices were correlated with each other and with the year (Table S2). We considered the correlations between these variables in our statistical approach, being aware of the hazards of multicollinearity (*Dormann et al., 2013*).

## Statistical analyses

We excluded the springs of 1993 and 2011, when fewer than 30 Willow Warblers were caught at Bukowo; thus the data for analyses included 34 years with at least 31 birds caught in a spring (Table S1). We imputed missing data for days when mistnetting was suspended for reasons such as heavy storms, as done in similar studies (*Sokolov et al., 1998*; *Tøttrup, Thorup & Rahbek, 2006a*; *Redlisiak, Remisiewicz & Nowakowski, 2018*). For a given date, missing values were imputed using the mean number of birds ringed on that date in the six years before and after the year with the missing value (*Redlisiak, Remisiewicz &*

*Nowakowski, 2018*). These occurred in two years and constituted at most four days per spring.

For each spring we calculated the annual accumulative curve, defined as the cumulative daily percentage of the numbers of Willow Warblers mistnetted that season, and used these curves to compare the timing of migration between years. We also computed the dates when 10% (beginning), 50% (median) and 90% (end) of migration occurred in each spring between 1982 and 2017. We calculated the duration of spring migration as the difference (in days) between the dates of 10% and 90% of migrant passage (*Miles et al., 2017*; *Lehikoinen et al., 2019*). To examine the long-term trend in the timing of Willow Warbler spring migration, we calculated the linear least square regressions of these percentiles and the duration against year (*Tøttrup, Thorup & Rahbek, 2006a*; *Miles et al., 2017*).

The date of each percentile reflects the timing for that stage of the spring passage, but does not reflect the overall migration pattern in a season. To quantify the extent to which the overall migration pattern in a particular spring was "early" or "late" in relation to the overall multi-year pattern, we developed the Annual Anomaly (AA), measured in days. We calculated the average cumulative curve based on the daily catches in all years. This provided the overall long-term arrival pattern, which served as the baseline (Fig. 2). We defined the AA for a year as the area between the cumulative curve for that year and the average curve (Fig. 2, Fig. S1), following *Crawford et al. (1990)* and *Nowakowski et al. (2005)*. We estimated the AA as the sum of the daily differences between the two curves over the 45 days of the spring migration (Fig. 2). This area can be positive or negative. A negative value for AA indicated that negative daily differences prevailed, and thus the passage in that year was earlier than the multi-year average pattern; a positive AA indicated the passage was later than the average (Fig. 2, Table S1, Fig. S1). This is analogous to studies analysing trends in the dates of percentiles of passage, where a negative trend indicates an advance of passage over the years and a positive trend means a delay (*Tøttrup, Thorup & Rahbek, 2006a*; *Miles et al., 2017*; *Lehikoinen et al., 2019*).

The linear regression of AA over the years describes the long-term trend in the timing of Willow Warblers' spring migration at the Polish coast, expressed as the integrated deviation from the long-term average pattern. To investigate the effect of climatic variation on the timing of the arrival of Willow Warblers in spring, we used the AA as the response variable in multiple linear regression models in which the year and the 13 climate variables which we chose (Table 1) were explanatory variables.

The AA and the climate indices were time series, so they potentially contained autocorrelations between successive values (*Cowpertwait & Metcalfe, 2009*; *Crawley, 2013*; *Simmons et al., 2015*; *Barshep et al., 2017*). The missing values of the AA (in 1993, 2011) were imputed for the autocorrelation tests, with the values predicted from their regression trend against the years (*Crawley, 2013*). We tested for any significance of autocorrelations in our time series for lags of up to 10 years by using a portmanteau Ljung–Box Q-test to test the null hypothesis of independence (a lack of autocorrelations) in the analysed data series (*Ljung & Box, 1978*). The autocorrelations were not significant for any variable (Table 1), so we were able to use standard regression analyses.

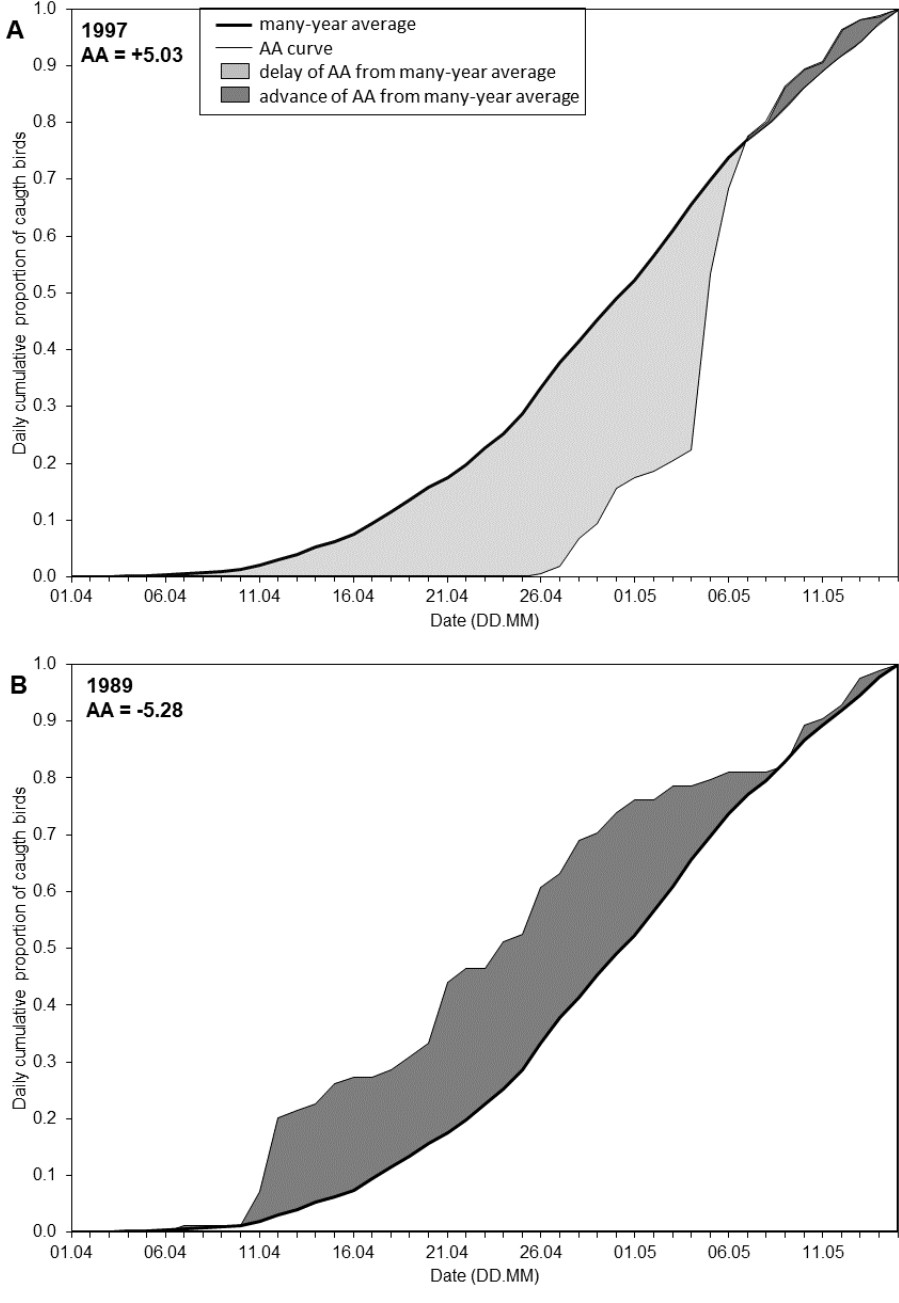

**Figure 2** **Visual representation of the Annual Anomaly (AA).** (A) Spring migration in 1997 in relation to the regression line of overall spring migration during 1982–2017; (B) Spring migration in 1989 in relation to the regression line of overall spring migration during 1982–2017 (Fig. 3B, Table S1).

We standardised the Annual Anomaly and the climate variables so that each had a mean of 0 and a standard deviation of 1. Then we checked for any correlations between the climate variables using Pearson's correlation coefficient (Table S2). All correlation coefficients but one were $|r| < 0.62$ (Table S2), which indicated no strong multicollinearity (*Dormann et al., 2013*). We retained the year as the explanatory variable in our model,

though it was correlated with a few climate variables (Table S2), to control for the effect of the year while estimating the effects of the climate variables on AA (*Frost, 2019*). To assess the multicollinearity of all the explanatory variables in the full and the best models we used the variance inflation factor (VIF) (*Dormann et al., 2013*; *Sparks & Tryjanowski, 2005*; *Marchowski et al., 2017*) in the R package "car 2.1-6" (*Fox & Weisberg, 2011*). Some studies have used NAOI as a quadratic term (*Møller, 2002*; *Grimm et al., 2015*), thus we compared models of AA regression against each climate variable used as a linear or as a quadratic term, using AICc, to determine which use would produce better models. After this comparison, we chose to use all climate variables in linear terms. We used multiple linear regression with the Annual Anomaly (AA) as the response variable with the year and the 13 climate indices as the explanatory variables to estimate any relationship between the indices and the timing of the spring passage of Willow Warblers at the Baltic Sea coast. We were aware of the issue of multicollinearity in our modelling and thus carefully observed the process and its outputs. We inspected the residuals to check if the full and the best models met the assumptions of the multiple regression (*Crawley, 2013*). Using "all subsets regression" we selected the best model by Akaike Information Criteria corrected for small sample size (AICc), using the package "MuMIn 1.43.6" (*Bartoń, 2019*). To verify the choice of the best model, we also used "stepwise backward" selection by AIC in the package "MASS 7.3-49" (*Venables & Ripley, 2002*). To check if the best model was not overfitted, we plotted residuals from the model with six variables against each remaining seventh variable (*Frost, 2019*). We estimated the proportion of variation explained by the best model using the adjusted coefficient of determination (AdjR$^2$). Using the package "heplots 1.3-3" (*Fox, Friendly & Monette, 2007*), we also calculated partial regression coefficients (partial $R^2$) in the multiple regression models. The partial $R^2$ estimates the proportion of variation explained by each variable in the model. The statistical analyses were conducted in R 3.4.4 (*R Core Team, 2018*).

## RESULTS

### Long-term trends in timing of Willow Warblers' spring migration at Baltic Sea coast

The timing of Willow Warblers' spring migration at Bukowo advanced by eight days for the start (10%) of the passage, by eight days for the median (50%) and by five days for the end (90%). All three of these were statistically significant (Fig. 3A, Table S4). Because the date for the first 10% of birds advanced more than the date for the last 90%, the duration of the spring passage of 80% of migrants (10%–90%) through Bukowo extended by six days on average over the 36 years of the study (Fig. 3A, Table S4). As measured by the Annual Anomaly (AA), spring migration advanced by 0.15 days a year ($t = 8.68$, $P = 0.006$), or 5.4 days over the 36 years (Fig. 3B, Table S4). Though significant, the linear trend explained only 32% of the variation in the starting dates of migration, 20% of variation in the median dates and 21% of the variation in AA (Table S4), which varied greatly between years (Fig. 3B, Table S1). The extremes in AA occurred in 1989 (seven days early relative to the regression line) and in 1997 (five days late) (Fig. 3B, Table S1).

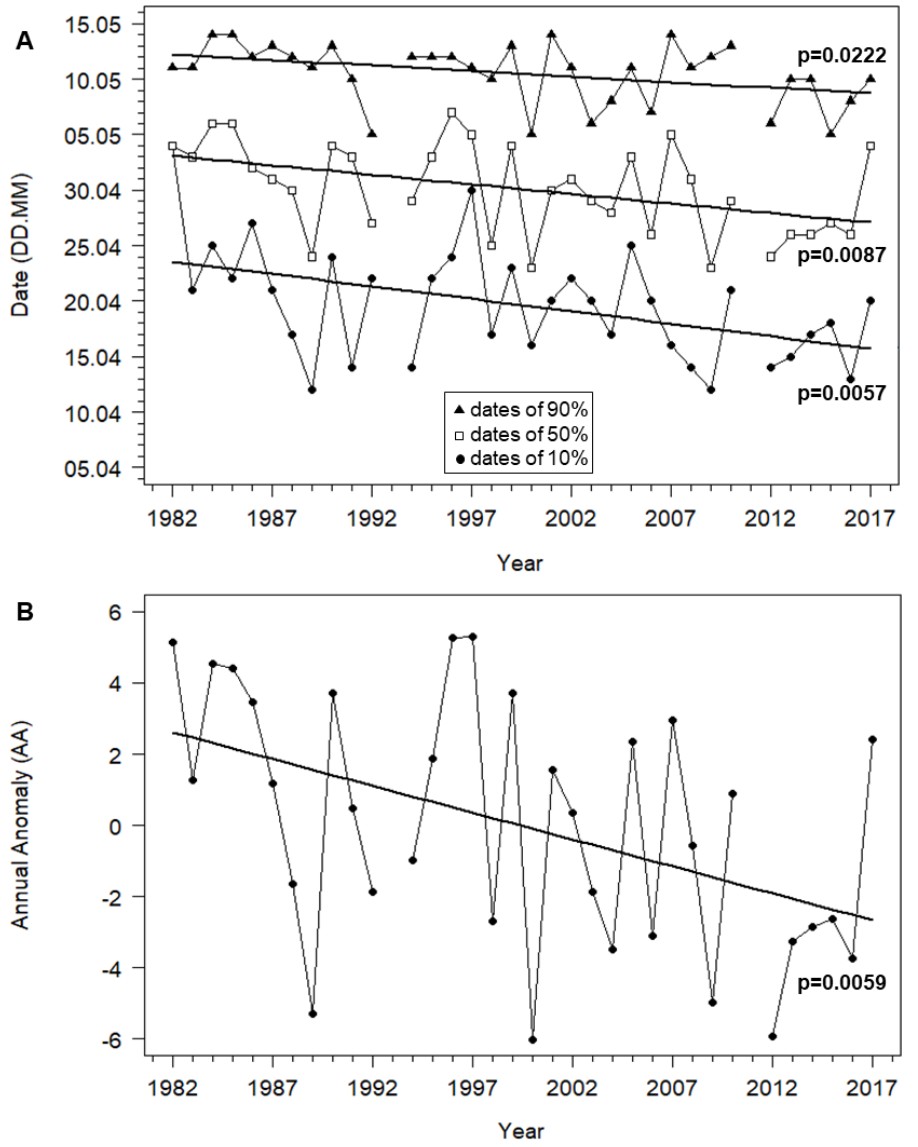

**Figure 3  Trends in migration dates of 10%, 50% and 90% and the Annual Anomaly (AA) for Willow Warbler spring migration at Bukowo, Poland, over 1982–2017.** (A) Dates of the beginning (10%), median (50%) and end (90%) of migration. (B) Annual Anomaly (AA) trends by linear regression. More statistics for regression equations in Table S4.

## The relationship between the timing of Willow Warblers' spring migration at the Baltic coast and large-scale climate indices

The full multiple regression model with AA as the response variable had 14 explanatory variables (13 climate variables and year) (Table 1 and Table S5). Values of VIF < 10 for all variables (Table S5) indicated no potentially harmful collinearity in our modelling (*Dormann et al., 2013*). We used all climate variables as linear terms, because for most the linear models of their relationships to AA were better than the quadratic models (Table S6). For NAOI in one (June–July) of four analysed periods the quadratic model was better

**Table 2   Pearson's correlation coefficients between the 13 climate indices and the year in 1982–2017 used in multiple regression models.**

| Explanatory variable | Estimate | SE | t | P | VIF | $R^2$ |
|---|---|---|---|---|---|---|
| LEB T Apr–May | −0.416 | 0.124 | −3.34 | 0.002 | 1.23 | 0.29 |
| NAOI Apr–May | −0.371 | 0.118 | −3.14 | 0.004 | 1.11 | 0.27 |
| NAOI Nov–Mar | −0.241 | 0.119 | −2.03 | 0.052 | 1.12 | 0.13 |
| SAH Nov–Mar | −0.242 | 0.116 | −2.08 | 0.047 | 1.08 | 0.14 |
| IOD Aug–Oct | −0.565 | 0.153 | −3.07 | <0.001 | 1.86 | 0.34 |
| SOI Aug–Oct | −0.385 | 0.149 | −2.58 | 0.016 | 1.78 | 0.19 |
| SCAND Jun–Jul | −0.396 | 0.133 | −2.98 | 0.006 | 1.41 | 0.25 |

Notes.

Significant correlations ($P < 0.05$) marked in bold face. VIF – variance inflation factors in the full model with all variables included. Abbreviations in Table 1.

than the linear one ($\Delta$AICc $< -2$; Table S6), thus we used NAOI as a linear term for all the periods for consistency. The best model with the smallest AICc value in the "all-subsets" approach (Table S7) had seven explanatory variables and explained Adj $R^2 = 58.6\%$ of the variation in the Annual Anomaly (Table 2). We obtained the same best model using the "stepwise backward" selection, which builds confidence that, given this set of candidate explanatory variables, we had selected a good subset for the final best model we present. Inspection of residuals (Figs. S2 and S3) showed that the best model and the full model met the assumptions of multiple linear regression, and the best model was well-fitted to the data (*Crawley, 2013*). Multicollinearity was not an issue in the best model because the largest variance inflation factor (VIF) was 1.86, substantially less than the guidance cut-off value of 10 (Table 2). SOI (August–October) explained 37% of the variation in the IOD (August–October), according to their linear regression (Table S8), thus these two indices largely varied independently though they were correlated (Table S2). The final best model had seven variables, based on 34 data points (Table 2). In these circumstances a single outlying value can result in an explanatory variable becoming significant. To confirm that overfitting was not an issue in this regression analysis, we performed seven regressions, each with six of the seven explanatory variables, and plotted the residuals from each regression against the missing explanatory variable (Fig. S4). In each case we obtained a well-behaved scatter plot showing a decreasing trend, with no outliers. In each plot it is clear that the inclusion of the missing explanatory variable is supported by the data (Fig. S4). Overall, the regression analysis can be characterised as satisfactory.

The year as a variable was not selected in the best-fitted model, indicating that the climate variables explained the multi-year trend in AA. The partial $R^2$ values ranged from 0.34 for the Indian Ocean Dipole (August–October) to 0.13 for the North Atlantic Oscillation Index (November–March) (Table 2). The signs of all the estimated regression coefficients were negative (Table 2). We can interpret these signs cautiously because most correlations between the seven variables included in the final model were not significant or relatively small (Table S2). When we accounted for the effects of other climate variables, the local April–May temperatures at Łeba were related to AA and migration advanced in warm springs (Table 2, Fig. 4A). Similarly, for each of the seven climate indices in the final model,

the higher the value of the index, the earlier was the timing of spring migration (Table 2, Fig. 4). The relationships of AA to the SOI (August–October) and SAH (November–March), and to temperature in Łeba (April–May), when each was a single explanatory variable, were not significant (Fig. 4). These relations were revealed in the multiple regression model, which controlled for the effects of the other variables on AA (Table 2).

## DISCUSSION

Two outcomes stand out in this study on the timing of Willow Warblers' spring migration at Bukowo on the Baltic Sea coast. The first outcome is novel: 59% of the interannual variation in the timing was related to seven large-scale climate indices on the species' wintering and breeding grounds over the previous 12 months. This suggests that the timing of Willow Warbler spring migration at the Baltic Sea coast is a summation of the ecological conditions which they had encountered over the previous year during breeding, migration south, wintering in Africa and migration north. We discuss the ways in which each of the seven climate indices could be related to spring migration at the Baltic Sea coast. Secondly, the study confirmed that a long-term trend for Willow Warblers to arrive earlier in spring, reported from other sites in Europe but analysed over shorter periods (*Sokolov et al., 1998*; *Cotton, 2003*; *Hüppop & Hüppop, 2003*; *Tøttrup, Thorup & Rahbek, 2006a*; *Hedlund et al., 2014*; *Miles et al., 2017*), continued at Bukowo up to 2017. We discuss the potential causes of these shifts in the context of our results and the earlier studies.

### Timing of Willow Warbler spring migration in relation to local conditions in spring

The earlier arrival of Willow Warblers in spring at Bukowo over 1982–2017 (Fig. 3) corresponds with the overall pattern for this species in Europe and shows that the advance in spring arrivals has continued. At the Baltic Sea island of Christiansø, Denmark, spring migration advanced in 1976–1997 by 0.25 days a year for the first 5% of birds and by 0.24 days a year for the first 50% (*Tøttrup, Thorup & Rahbek, 2006a*); the equivalent values in our study were 0.27 and 0.17 days a year, and 0.22 for the first 10% of birds, a more robust metric than 5%, considering our small samples (Fig. 3, Table S4). On the island of Gotland, Sweden, spring arrival of Willow Warblers advanced by 0.20–0.29 days a year and egg-laying by 0.23 days a year in 1990–2012 (*Hedlund et al., 2014*). Spring migration of Willow Warblers also advanced in 1959–1990 at Rybachy, Russia, ca 220 km east of Bukowo on the Baltic coast (*Sokolov et al., 1998*), in 1960–2002 at Helgoland, North Sea, Germany (*Hüppop & Hüppop, 2003*), in 1955–2014 at Fair Isle, Scotland (*Miles et al., 2017*), and in 1971–2000 in Oxfordshire, England (*Cotton, 2003*).

We found an increase in the duration of Willow Warblers' spring migration as a result of their passage starting earlier; this corresponds with similar patterns reported by *Miles et al. (2017)* and *Lehikoinen et al. (2019)* from other locations. The pattern of Willow Warblers' earlier arrival in spring at Bukowo was related to local mean April–May temperatures in Łeba (Fig. 4A), which increased over the 36 years of our study (Table S3). Our results correspond with other findings: the early arrivals of Willow Warblers at Rybachy and in Estonia were associated with high local temperatures in April and May (*Sokolov et al.,*

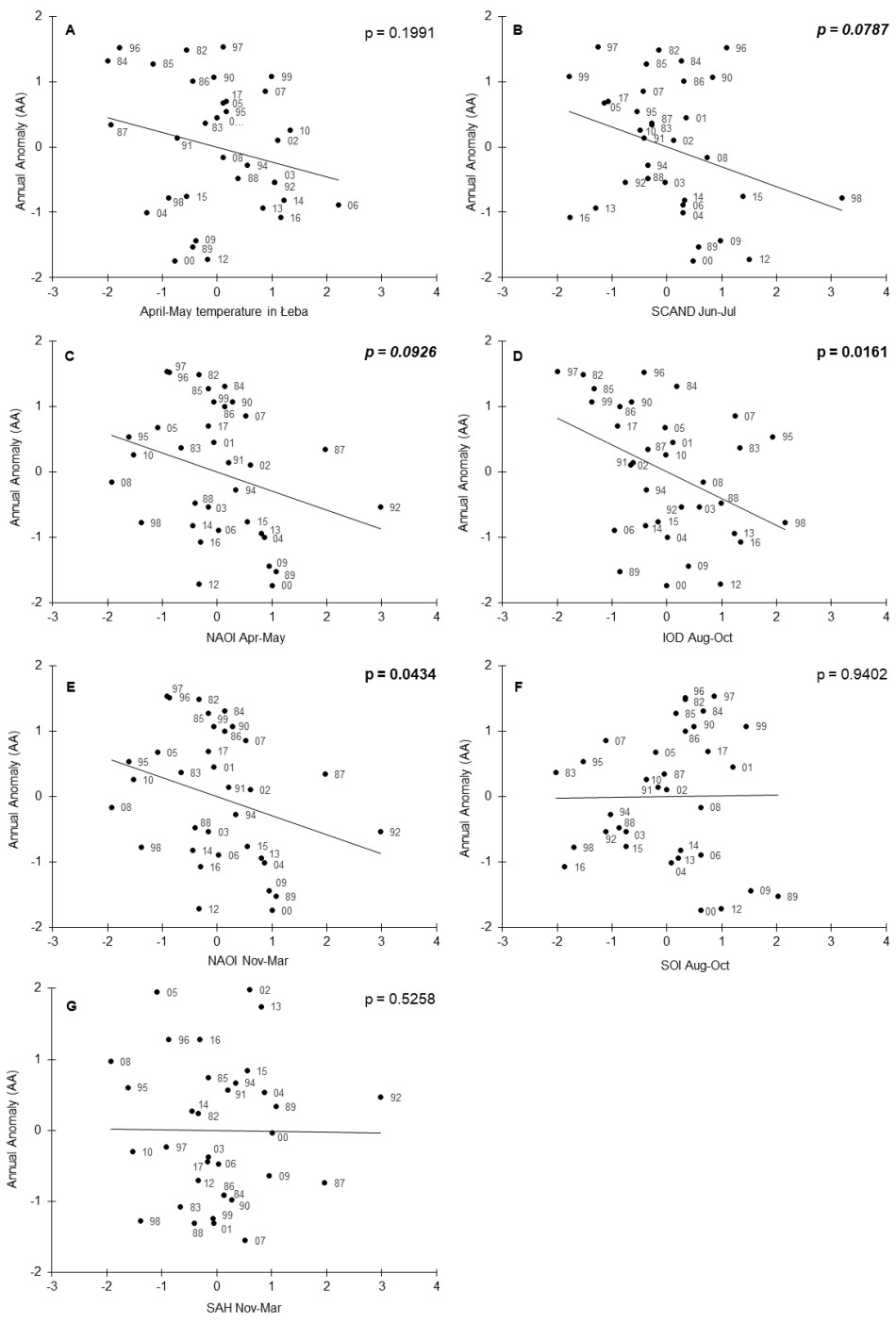

**Figure 4** **Annual Anomaly (AA) of Willow Warblers' spring migration at Bukowo in 1982–2017 against the climate indices in the best model (Table 2).** All variables presented as standardised values. The numbers alongside the black circles are the last two digits of the year. $p < 0.05$ marked in bold face, $0.05 < p < 0.1$ in italics. (A) AA against mean of local temperature in Łeba in April–May. (B) AA against mean Scandinavian Index in June–July. (C) AA against mean North Atlantic Oscillation Index in April–May. (D) AA against mean Indian Ocean Dipole in August–October. (E) AA against mean North Atlantic Oscillation Index in November–March. (F) AA against mean Southern Oscillation Index in August–October. (G) AA against mean Sahel Precipitation Anomaly in August–October.

*1998*; *Palm et al., 2009*). At Helgoland the earlier migration of Willow Warblers in spring coincided with higher mean annual temperatures locally (*Hüppop & Hüppop, 2003*). However, these correlations between the early arrival of Willow Warblers and high local temperatures at ringing stations might not be a cause–effect relationship but rather the effect of both phenomena being related to high temperatures at an earlier stage of the birds' migration. A strong correlation between the dates of spring arrival of the first 5% of Willow Warblers' at Hanko, Finland, and mean April temperatures in the Balkans and in the Middle East supports this premise (*Halkka, Lehikoinen & Velmala, 2011*). The advance of Willow Warblers' spring migration corresponds with a general trend in Europe over the past few decades for migrants to arrive earlier in relation to global warming and the resulting higher spring temperatures along the birds' migration routes (e.g., *Sokolov et al., 1998*; *Tryjanowski, Sparks & Kuźniak, 2002*; *Cotton, 2003*; *Hüppop & Hüppop, 2003*; *Ahola et al., 2004*; *Tøttrup, Thorup & Rahbek, 2006a*; *Saino et al., 2007*; *Miles et al., 2017*; *Lehikoinen et al., 2019*). April–May temperatures at Łeba explained only 29% of the variation in Willow Warbler spring migration at Bukowo, but these temperatures and six large-scale climate indices together explained 59% of the interannual variation (Table 2).

The average April–May NAOI explained 27% of the variation in the timing of spring migration (Table 2, Fig. 4C); this contrasts with *Palm et al. (2009)* who found no relation between the first arrival dates of Willow Warbler in Estonia in 1957–1996 and March–May NAOI. But our results come from later years, when the effects of climate change have been more pronounced (*Simmons et al., 2017*). Additionally, Willow Warblers begin arriving in the Baltic region from April, thus the April–May NAOI which we used probably reflects conditions during spring migration through Europe better than March–May. The April–May NAOI was not correlated with mean April–May temperatures in Łeba (Table S2); thus each of these climate variables explains a different aspect of the variation in the timing of spring migration at Bukowo. In warm springs, with high April–May temperatures in Europe and high NAOI, early availability of insects on passage would enable migrants to fuel quickly at stopovers, migrate faster and arrive earlier at the breeding grounds (*Van Noordwijk et al., 2006*; *Gordo, 2007*). Early-arriving individuals occupy better-quality territories and have better breeding success than birds which arrive later; thus there is selective pressure for early migration in warm and early springs (*Kokko et al., 2006*; *Gordo, 2007*).

### Interannual variation in timing of Willow Warblers' spring migration in relation to climate variability in west, east and southern Africa

In the tropics and sub-tropics rainfall has a greater effect on food abundance for birds than temperature, in contrast to the temperate zones of the northern hemisphere (*Elkins, 1988*; *Newton, 2010*). NAOI, SAH, SOI and IOD are related to rainfall and temperatures in Africa, and thus with vegetation and ecological conditions on the migrants' passage and wintering areas (e.g., *Hurrell, 1995*; *Stenseth et al., 2003*; *Marchant et al., 2006*; *Munemoto & Tachibana, 2012*; *Heino et al., 2018*). Ecological conditions at the wintering grounds determine the body condition of birds, influence their timing of moult and fuelling, and thus the timing of departure and the rate of migration north (*Katti & Price, 1999*;

*Salewski et al., 2004*; *González-Prieto & Hobson, 2013*; *McKellar et al., 2013*; *Ouwehand & Both, 2017*; *Remisiewicz et al., 2019*). Wet winters in Africa produce an abundance of insects (*Allan et al., 1997*; *Lingbeek et al., 2017*; *Thorup et al., 2017*) and enable migrants to fuel for migration quickly and to depart early (*Peach, Baillie & Underhill, 1991*; *Studds & Marra, 2011*; *Altwegg et al., 2012*). Abundant food also enables the migrants to moult early, leaving them time for efficient pre-migratory fuelling (*Salewski et al., 2004*; *Remisiewicz et al., 2019*). Greenish Leaf Warblers *Phylloscopus trochiloides* fuelled faster, accumulated greater fuel loads and departed earlier from their wintering grounds in tropical India in wet years, when insects were more abundant, than in dry years (*Katti & Price, 1999*). We suggest a similar pattern for Willow Warblers wintering in Africa. Large fuel reserves on departure from the wintering grounds enable birds to migrate faster and with shorter stopovers (*Gordo, 2007*; *Zwarts et al., 2009*; *McKellar et al., 2013*). In contrast, droughts at the wintering and stopover sites in Africa delay the arrival in the European spring of long-distance migrants such as Red-backed Shrike and Thrush Nightingale *Luscinia luscinia* (*Tøttrup et al., 2012*). Temperatures at the wintering grounds might also influence food availability and thus the migration timing of birds. Willow Warblers and other long-distance migrants arrived early in Oxfordshire, UK, in spring after high temperatures during their wintering in Africa (*Cotton, 2003*). The breeding success of female White Storks in western Poland showed a relationship to climate indices which reflect rainfall in west Africa (SAH) and in east Africa (IOD) in August–September. Different storks use one or the other of these winter quarters (*Tobolka et al., 2018*). We analysed climate indices for more areas across Africa and Europe occupied by our study species at different life stages; thus we found a more complex combination of climate indices to be related to their spring arrival in Poland than *Tobolka et al. (2018)* revealed for White Storks. The large-scale climate indices reflect rainfall and temperatures in regions of Africa and Europe used by different populations of Willow Warblers. We suggest that the relationships we found between these indices and spring migration phenology of Willow Warblers at the Baltic Sea coast reflect carry-over effects of ecological conditions which the different populations encountered at locations they visited in the preceding year.

The relation we found between November–March NAOI and Willow Warbler spring migration (Table 2, Fig. 4E) corresponds with the early arrival of many species in Europe after a positive winter NAOI (December–February), related to warm and early springs (e.g., *Forchhammer, Post & Stenseth, 2002*; *Hüppop & Hüppop, 2003*; *Ahola et al., 2004*; *Vähätalo et al., 2004*; *Rainio et al., 2006*; *Gordo, 2007*; *Jonzén et al., 2007*; *Saino & Ambrosini, 2008*; *Halupka & Halupka, 2017*). At Helgoland, several passerines begun arriving early after a positive December–March NAOI (*Hüppop & Hüppop, 2003*), though NAOI explained only a small part of the variation in spring migration phenology in short- and long-distance migrants (*Haest, Hüppop & Bairlein, 2018*). Similarly, in our study, November–March NAOI explained only 13% of the variation in the timing of Willow Warblers' spring migration (Table 2). November–March NAOI was correlated with the April–May mean of local precipitation in Łeba during our study period (Table S2); we did not include precipitation as a factor in our models because we focused on large-scale climate variables. However, this correlation suggests that the timing of birds' spring migration might be

related to winter NAOI indirectly. The direct factor which influences migratory birds' timing might be rather the local rainfall in spring, which is related to winter NAOI. Winter NAOI had a stronger effect on the arrival of short- and medium-distance migrants in Norway than on long-distance migrants (*Forchhammer, Post & Stenseth, 2002*). Long-distance migrants breeding in central and northeastern Europe, including Willow Warbler, migrate farther east and south in Africa than the short- and medium-distance migrants (*Cepák et al., 2008*; *Fransson & Hall-Karlsson, 2008*; *Bairlein et al., 2014*; *Valkama et al., 2014*). Our results provide evidence that other large-scale climate indices, which reflect conditions in east and southern Africa, help explain interannual variation in the timing of spring migration in long-distance migrants better than NAOI alone (Table 2).

Of all the climate indices we analysed, the Indian Ocean Dipole (August–October) explained the greatest part (39%) of interannual variation in the timing of Willow Warblers' spring migration (Table 2, Fig. 4D). This supports the suggestion by *Tryjanowski, Stenseth & Matysioková (2013)* that the IOD is an important indicator of climate conditions which influence birds breeding in Europe. Red-breasted Flycatchers *Ficedula parva* arrived early in spring in Białowieża Forest, eastern Poland, in years with a high November–March IOD (*Mitrus, Sparks & Tryjanowski, 2005*; *Tryjanowski, Stenseth & Matysioková, 2013*). We found a similar relationship between Willow Warblers' arrivals and August–October IOD (Fig. 4D). The timing of the first clutches of Red-backed Shrikes in central Europe was related to winter IOD, probably through the positive effect of high rainfall in east Africa on their departure condition (*Hušek et al., 2008*). Female White Storks laid more eggs in western Poland following a high IOD in August and September, but the timing of their spring arrival was not related to the IOD (*Tobolka et al., 2018*). In the light of these studies, we suggest that a positive August–October IOD, related to high rainfall and temperatures over August–December in east and southeast Africa, has positive effect on the condition of Willow Warblers using these areas for wintering and stopovers, which would facilitate their early arrival in spring at Bukowo. In contrast, a negative winter IOD, indicating low rainfall and temperatures in east and southeast Africa, might be related to the delayed spring arrival of Willow Warblers in Europe, as in other long-distance migrants (*Tøttrup et al., 2012*).

The relationship we found between August–October SOI and the timing of Willow Warblers' spring migration (Table 2, Fig. 4) corresponds with an analogous, though non-significant, relationship between the winter SOI (December–February) and the arrival of long-distance migrants in Oxfordshire, UK (*Cotton, 2003*). Willow Warblers, and other long-distance migrants, arrived early in Oxfordshire after high mean winter (December–February) temperature anomalies in Africa south of 20°N (*Cotton, 2003*). This could be related to a high winter NAOI, IOD and SOI (*Stenseth et al., 2003*; *Marchant et al., 2006*; *Schott, Xie & McCreary, 2009*); the three climate indices which were related to this species' arrival. Our results on SOI correspond with those of *Miller-Rushing et al. (2008)*, who showed that several medium-distance migrants arrived early in spring in Massachusetts, USA, after winters with a positive SOI, related to high rainfall in Central America and the Caribbean, where those species winter. The effects of the IOD and SOI on the timing of Willow Warblers' spring passage suggest that a considerable proportion of the population

which migrates through the Baltic coasts use east and southeast Africa as stopover and wintering grounds, concurring with migratory connections shown by ringing recoveries (*Fransson & Hall-Karlsson, 2008*; *Zwarts et al., 2009*; *Bairlein et al., 2014*; *Valkama et al., 2014*; *Maciąg et al., 2017*).

The relationship we revealed between the Sahel index (November–March) and the timing of Willow Warblers' spring migration at Bukowo (Table 2, Fig. 4G) corresponds with the finding that in the spring of dry years Willow Warblers suffered twice the mortality crossing the Sahara Desert than in wet years (*Zwarts et al., 2009*). As with Willow Warblers, Barn Swallows arrived in Spain early in the spring after winters with a high SAH in west Africa (*Gordo, 2007*). Several passerines, including Willow Warblers, arrived early in spring at Capri, Italy, after wet and hot winters in north Africa and the Sahel (*Saino et al., 2007*). Female White Storks in western Poland laid more eggs after a high SAH in August and September (*Tobolka et al., 2018*). In the context of these studies, we suggest that good rainfall in the Sahel during Willow Warblers' stopovers, reflected by a high SAH, probably has a positive effect on their condition and thus facilitates their early arrival at Bukowo in spring.

## Carry-over effects of conditions at earlier life stages on Willow Warbler spring migration

The relationship of Willow Warbler spring migration to IOD and SOI in August–October (Table 2, Figs. 4D and 4E), rather than in November–March when they overwinter in Africa, can be explained in two ways. Firstly, the August–October IOD and SOI influence the ecological conditions which the birds encounter when they arrive in east and southern Africa a few months later. Positive IOD and SOI are related to high rainfall in east and southeast Africa in October–December (*Marchant et al., 2006*), which has a positive effect on vegetation in later months (*Heino et al., 2018*); this would increase the food available for the birds. Secondly, the conditions Willow Warblers encounter during their August–October migration south across east and southeast Africa might have carry-over effects on the next stages of their lives, as in other species. For example, *Aloni, Markman & Ziv (2019)* showed that high temperatures in September–November in north and east Africa, which likely increase insect abundance, when the insectivorous Bonelli's Warbler *Phylloscopus orientalis* and Lesser Whitethroat *Sylvia curruca* migrate south through the area, were related to their good body condition and early arrival in spring (February–May) at a stopover site in Eilat, Israel. Analogously, in the European Reed Warbler low rainfall at autumn stopover sites in Spain and Morocco had carry-over effects on their reduced survival and later return to the breeding grounds the following spring (*Halupka et al., 2017*). The earlier arrival and increased brood size of Redstarts *Phoenicurus phoenicurus* in the UK was related to high rainfall in the Sahel (*Finch et al., 2014*), and the high number of eggs laid by White Storks in Poland was related to high positive SAH and IOD in August and September (*Tobolka et al., 2018*). Early arrival of American Redstarts *Setophaga rucilla* at high-quality wintering sites had a carry-over effect in their early arrival at their breeding sites (*McKellar et al., 2013*). These findings support our reasoning, and both explanations are likely to be correct for Willow Warblers.

We suggest that the earliest and the latest spring migrations of Willow Warblers at the Baltic coast reflect the combined effects of the conditions the birds encounter at different life stages at multiple locations. The outstandingly early migration of Willow Warblers in 1989 coincided with a positive November–March and April–May NAOI (Figs. 4C and 4E), related to an early and warm spring in Europe (*Hurrell, 1995*) and dry spring locally at Łeba (Table S2), and with a high positive August–October SOI (Fig. 4F). Such high SOI was related to a strong La Niña anomaly in 1988–1989 (*Null, 2019*), which brought abundant rainfall to southeast Africa (*Stenseth et al., 2003*). The outstandingly late migration in 1997 coincided with the lowest August–October IOD of all the years we studied (Fig. 4D), indicating low rainfall in east and southeast Africa (*Marchant et al., 2006*), and with a negative November–March and April–May NAOI and June–July SCAND index (Figs. 4C and 4E). Globally, 2011–2016 were the warmest five years in the past century (*Simmons et al., 2017*). Willow Warblers arrived at Bukowo early in the spring of 2012–2016 (Fig. 2, Table S1), and high local spring temperatures (Fig. 4A) might be the proximate factor facilitating early arrivals, through the local availability of ample food. However, our results suggest that conditions on the wintering grounds made a decisive contribution to the pattern of Willow Warbler arrivals. The IOD (August–October) in 2012, 2013 and 2016, and SAH (November–March) in 2013 and 2015 were positive (Figs. 4D and 4G), indicating good rainfall in east and west Africa. But El Niño in 2015–2016 caused a widespread drought in southern Africa (*Heino et al., 2018*), reflected by a negative August–October SOI (Fig. 4F). The following April–May 2016 and 2017, Willow Warblers departed from South Africa later than in 2008–2015 (http://sabap2.birdmap.africa/), as we would expect if drought had weakened their condition. However, Willow Warblers which wintered in western or eastern Africa might have benefitted from high rainfall there, which would probably facilitate the overall early arrival at Bukowo in 2015 and 2016. In contrast, the late spring migration of Willow Warblers at Bukowo in 2017 (Fig. 2), despite a warm April–May at Łeba (Fig. 4A), might be related to a combination of low August–October IOD and low June–July SCAND (Figs. 4D and 4B). Our results suggest that the conditions the long-distance migrants encounter more than half a year earlier and far away might be more crucial drivers for their early or late arrivals at the breeding grounds in spring than the conditions they experience on spring migration.

## Annual anomaly as an index of phenological events

The shifts in the migration phenology of birds have often been examined using first occurrence dates (FAD) (e.g., *Tryjanowski, Sparks & Kuźniak, 2002*; *Cotton, 2003*; *Mitrus, Sparks & Tryjanowski, 2005*; *Rubolini et al., 2007*; *Mason, 2009*; *Usui, Butchart & Phillimore, 2017*), which might be biased by observer activity, the population size and the presence of outstandingly early individuals (*Tryjanowski & Sparks, 2001*; *Tryjanowski, Kuźniak & Sparks, 2005*; *Gordo, 2007*), for "one swallow does not a summer make" (Aristotle 384-322BC, *Etica Nicomachea*). Other measures used to analyse the timing of passage in birds are the mean dates of migration (e.g., *Sokolov et al., 1998*; *Hüppop & Hüppop, 2003*; *Usui, Butchart & Phillimore, 2017*; *Haest et al., 2019*) and the dates of percentiles of passage, most often of the beginning (5% or 10%), median (50%), and end (90% or 95%),

which reflect different stages of a species' migration (*Tøttrup, Thorup & Rahbek, 2006a*; *Lehikoinen, Santaharju & Møller, 2017*; *Miles et al., 2017*; *Aloni, Markman & Ziv, 2019*). These are all useful metrics, but none reflects the pattern of the entire migration. AA is one value which is correlated with the dates of all percentiles, but has a smaller variance than the dates of most percentiles, except for 90% and 95% (Table S9). We encourage the use of AA as a convenient single measure which reflects the pattern of a species' migration at a location in relation to the many-year average. We calculate AA in a similar manner to a temperature anomaly, which is a departure from a long-term baseline, for example the monthly temperature in a location averaged over a century. Temperature anomalies enable comparisons between locations with different baselines and facilitate studies of climate change (*US National Oceanic and Atmospheric Administration, 2019*). Analogously, the AA can also be derived from any cumulative data on biodiversity which have a long-term baseline. The Annual Anomaly of bird migration would be easy to derive, update annually and compare, based on existing datasets and continuing activities of bird observatories (e.g., *Sokolov et al., 1998*; *Hüppop & Hüppop, 2003*; *Tøttrup, Thorup & Rahbek, 2006b*; *Miles et al., 2017*; *Aloni, Markman & Ziv, 2019*; *Lehikoinen et al., 2019*).

## CONCLUSIONS

We provide evidence of the combined effect of large-scale climate indices which operate across Africa and Europe on the phenology of spring migration of a long-distance migrant passerine in Europe. The results support our initial assumption that Willow Warblers caught at Bukowo use wintering grounds which span west, east and southeast Africa. We suggest that a combination of large-scale climate indices, such as NAOI, SAH, IOD and SOI, related to conditions at the non-breeding grounds, along with indices such as SCAND, related to conditions at the breeding grounds, probably reflect ecological drivers for phenological changes in birds with complex patterns of migrations between continents. Our results emphasise that understanding the effects of climate change on the migration phenology of a migratory bird in one location requires examination of carry-over effects of climatic variation which the different populations experienced at previous life stages in multiple locations. We recommend the use of the Annual Anomaly metric to facilitate studies of long-term phenological responses of birds, and other organisms, to climate change at different locations.

## ACKNOWLEDGEMENTS

The citizen scientists who collected the data at Operation Baltic's Bukowo ringing station made a decisive contribution to this paper. Eva Poślińska compiled the data for climate indices and helped write R scripts. Michał Redlisiak and Joanna Gruchocka compiled data for the Łeba weather station from http://www.ecad.eu/. The comments from Scott Edwards and three anonymous reviewers helped to improve the manuscript. Joel Avni commented on and edited earlier drafts.

### Funding

This study was supported by a research grant from the National Research Foundation, South Africa, and the National Centre for Research and Development, Poland, within the Poland-South Africa Agreement on Science and Technology (PL-RPA/BEW/01/2016). Collation and digitisation of data were supported by Special Research Facility grants (SPUB) of the Polish Ministry of Science and Higher Education to the Bird Migration Research Station, University of Gdańsk. The funders had no role in study design, data collection and analysis, decision to publish, or preparation of the manuscript.

### Grant Disclosures

The following grant information was disclosed by the authors:
National Research Foundation, South Africa.
National Centre for Research and Development, Poland.
Poland-South Africa Agreement on Science and Technology: PL-RPA/BEW/01/2016.
Special Research Facility grants (SPUB) of the Polish Ministry of Science and Higher Education to the Bird Migration Research Station, University of Gdańsk.

### Competing Interests

The authors declare there are no competing interests.

### Author Contributions

- Magdalena Remisiewicz conceived and designed the experiments, performed the experiments, analyzed the data, prepared figures and/or tables, authored or reviewed drafts of the paper, and approved the final draft.
- Les G. Underhill conceived and designed the experiments, performed the experiments, analyzed the data, authored or reviewed drafts of the paper, and approved the final draft.

### Animal Ethics

The following information was supplied relating to ethical approvals (i.e., approving body and any reference numbers):

Catching and ringing birds was conducted with the permission from the Polish Academy of Sciences, with the approval the General Directorate for Environmental Protection, Poland (DZP-WG.6401.03.97.2017.jro).

### Field Study Permissions

The following information was supplied relating to field study approvals (i.e., approving body and any reference numbers):

Field research at Bukowo was approved by the Marine Office, Słupsk (OW-A-510/87/17/ds).

### Data Availability

The raw data are available in the Supplemental Files.

## Supplemental Information

Supplemental information for this article can be found online at http://dx.doi.org/10.7717/peerj.8770#supplemental-information.

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
