# Peer review of "Climatic variation in Africa and Europe has combined effects on timing of spring migration in a long-distance migrant Willow Warbler Phylloscopus trochilus"

_PeerJ, doi:10.7717/peerj.8770_

## Round 0.1 · original submission · Minor Revisions

· Academic Editor

Minor Revisions

All three reviewers felt the manuscript presented novel data and comprehensive analyses. Although reviewer 1 presented a number of comments on your paper, I regard this review still as a minor revision, although some of the recommended revisions might involve additional data analysis. In particular this reviewer asked you to look in potential correlations between climate variables and different patterns of migration responses to climatic indexes. The other two reviewers had fairly minor suggestions about interpretation, reporting and citations.

Reviewer 1 ·

Basic reporting

The English language should be improved in several places to ensure that an international audience can clearly understand your text. Some examples where the language could be improved are given below. The current phrasing, particularly in the field methods section, makes comprehension difficult.
The introduction should show a stronger red line with transitions between facts.

The manuscript includes sufficient introduction and background and demonstrates how the work fits into the broader field of knowledge. Relevant prior literature has been appropriately referenced. The introduction has parts with excessive citations and the selection of the references cited is rather arbitrary. The discussion is excellent.

The structure of the article conforms to an acceptable format of ‘standard sections’.
Regarding the figures, I would suggest to exchange Figures 3 and S3 since Fig. S3 seems more important to a broader audience than Fig. 3.
Figures in the text are appropriately labelled. All appropriate raw data have been made available.
Figure S1 should, if possible, be included in the main text as it contains interesting and relevant information necessary to calculate the indices.
Figures S2 and S3 need a heading and a very short description. Likewise, Tables S1-S4 need a caption and/or a short description of what is shown. Specifically: Table S2: What has the Chi-square test been used for?

The submission is ‘self-contained,’ represents an appropriate ‘unit of publication’, and includes all results relevant to the hypothesis.

Experimental design

The submission is original primary research and clearly defined the research question, which is highly relevant and meaningful. The knowledge gap being investigated has been clearly identified, and statements has been made as to how the study contributes to filling that gap.

The investigation has been conducted rigorously and to a high technical standard. The research has been conducted in conformity with the prevailing ethical standards in the field. However, please see my concerns in parts of the statistical methods (described below).

Methods have been described with sufficient information to be reproducible by another investigator. All method descriptions are very comprehensive and allow a full understanding of the field and statistical methods. I am confident that the authors have a comprehensive dataset that is well suited to answer their questions

Validity of the findings

This study will have a high impact in the field and explores the novelty of examining several climatic indices on various scales at the same time.

The data on which the conclusions are based have been provided. The data seem to be robust, statistically sound, and controlled.

The conclusions have been appropriately stated, are connected to the original question investigated, and are be limited to those supported by the results.

Additional comments

General comments:
The authors tried to relate the arrival of the willow warbler to different local and distant climates. The topic is highly interesting and results will contribute substantially to the understanding of a widely observed phenomenon. I commend the authors for their extensive data set, compiled over many years of detailed fieldwork. I generally also like the idea of the calculation of the AA value, especially regarding the options of comparability among other regions or species.
However, I have some serious statistical concerns that I described in detail below. Main points include the model selection strategy, the way and time collinearity has been assessed, the absence of precipitation considerations, the relation of number of observations to number of parameters (n = 34, k = 14), and model reporting. I am giving examples and advices including literature for further read below.
Furthermore, the English language should be improved in several places to ensure that an international audience can clearly understand your text. Some examples where the language could be improved are given below. The current phrasing, particularly in the field methods section, makes comprehension difficult.
Abstract:
The abstract is well written, very informative and makes the reader curious. I would probably delete “(minimum AIC)” after the best model as I believe that this is common knowledge and if not, it can be read in the methods part. (see my concerns about the use of AIC)

Introduction:
While this section is very comprehensive, some parts rather read like a concatenation of facts than a story (in contrast to the discussion, which has a better storyline). The authors may want to consider better transitions among the facts to allow an improved story flow. Also, I would suggest to avoid excessive citations (see below).
Lines 50-52, 53-56 and 58-60: Such excessive citations make it hard for readers; Moreover, there are many more papers investigating migratory birds and timing, or response to NAO; I would suggest to cite a few that deal with many species and write “e.g.” before; otherwise the selection of the paper cited would seem a bit arbitrary.
Line 57: not only winter, also annual indices have been used (e.g., Møller 2002, Grimm-Seyfarth et al. 2015)
Line 61: “explained” instead of “explains”
Lines 106-109: It seems that this formulation is a bit misleading. My suggestion would sth. like: “We aimed to examine whether the time of spring migration of the willow warbler at the Baltic Sea coast had responded to large-scale climate variables affecting the wide non-breeding grounds in the year before migration.”

Materials & Methods:
This section is again very comprehensive and allow a full understanding of the field and statistical methods. I am confident that the authors have a comprehensive dataset that is well suited to answer their questions. However, the English language should be improved in some parts of the section to ensure that an international audience can clearly understand your text. Some examples are given below. The current phrasing makes comprehension difficult.
The statistical part is particularly well described and not many language improvements are necessary here. It is written in clear and understandable language. However, I have a few serious questions and concerns regarding the statistics:
(1) My first question is why the authors focused local climate on temperature only. First, the climate indices also represent various climatic factors and not just temperature. Second, rainfall, and even the interaction between temperature and rainfall, might have a strong influence on the appearance of the birds as they influence both their habitat and their food. If data are available for the weather station, I would encourage the authors to include rainfall and temperature-rainfall-interaction into the parameter set.

(2) A second question that came to my mind is whether all those climate indices could be correlated with each other. A statement on correlation of the values of the indices used in the model would help here. In general, it can be quite critical to use a multiple regression model with several indices that could potentially affect each other (and also local temperature or the “year” variable), even if a correlation is not directly detectable. The authors should make clear that this does not violate any model assumptions. (NB: See my detailed comments below; I figured that come tests have been conducted, but AFTER model selection, but model assumptions need to be checked BEFORE model selection)

(3) The authors only considered linear effects of the indices, while others have already shown that indices can have a quadratic influence (e.g., Møller 2002, Grimm-Seyfarth et al. 2015 for NAO). To my understanding, migration conditions might either become optimal toward one side of an index (linear relation) or around a specific index value and might get worse both above and below it (quadratic relation). A simple AICc comparison of AA against an index linearly or quadratically would solve this question.

(4) I have some substantial remarks regarding the model selection, reporting and model fit sections which I describe below in details. Particularly, the confusion of the authors between “all-subset regressions” and “backward selection” need to be clarified. See below.

(5) A LM with 34 observations and 14 explanatory parameters is extremely challenging and can easily overfit the model. I am not sure whether the use of “date of first capture per individual per year” as a response in a mixed model (using “Year” as random intercept) would not be more appropriate instead of using the AA value, even if I like the idea of the AA value as a nice single value that describes the behavior of the whole population. In addition, adding rainfall and temperature-rainfall interaction would add two more parameters. A possible alternative approach would be the following:

a. Choose parameter subsets, for example regarding the season or regarding the index (e.g., all values from spring, all from summer, all from winter; or all NAOIs, all SOIs etc.; no index doubling among subsets)
b. Conduct model selection on all possible parameter combinations per subset (NOT backward selection)
c. Put the “best parameters” per subset into a final “global model” and perform model selection on all possible parameter combinations

Specific comments:
Lines 113-115: This sentence needs some clarifications: I guess the information is available at the Baltic Sea cost? So is the Baltic Sea the final breeding ground or another stopover area? If first, I would suggest to remove “to northern Europe”, if the latter, I would suggest to add “at the stopover site xxx”. (NB: I realized it is the latter when reading the text further, but the first sentence is confusing).
Line 119-120: This sentence is again confusing
L126: please add: in Africa
L128: replace “so we use” by “which is why we use”
Climate indices section: The climate indices are well described und understandable even when not working in the subject area. However, I would suggest to either mentioning them in the beginning (but this will add a bit of text – despite the abbreviations could be used when explaining their effects further on), or show them in a table (e.g., Name, Abbreviation, part of Africa or Europe they have an influence on). This could make it easier to follow (and to have a look-up when continuing in the text).
L 173: Could the authors please explain whether the temperatures from Leba represent temperatures in Bukowo? Has there ever been comparative measurements? Or does it experience the same geographic conditions? The distance of 95 km sounds rather far.
Line 174-176: Could the authors give a brief overview on the behavior of temperature across the 40 years? Where they increasing and if so, how much on average or how much per year?
Lines 182-185: Where the averaged indices NAOI April-May and NAOI November-March and NAOI August-October correlated with each other? It is a rather “slow responding” index so I could imagine it does not change much from summer to winter or from winter to spring. The same holds for other indices averaged across several seasons. A few words (if or if not, and if so, what have the authors done) would help here.
Lines 209-210: The method used is very valid and known, I believe that the two most important references are sufficient.
Line 233: The authors describe that the year and the 13 climate variables were used in the same LMs. Did the authors test for correlations of the indices and the year as I would assume that at least temperature could be related to the year, but even indices could advance or delay over time.
Line 237-238: Even if it is convincing to read that the authors know what they would need to do in the case of autocorrelation, if is not necessary to state this here, as the autocorrelations were not significant in your case.
Lines 245-246: This part is extremely important and I am waiting for it since the description of the climate data. I would suggest to move it to the end of the climatic section. That way, the question about potential collinearity is explained earlier. Moreover, I am missing the results for the collinearity analyses which are needed before heading on to the LM description.
Line 249: The authors claim to use “all subsets regressions”, which is not true if they also state that they used stepwise backward selection. I have had a look at the R code and figured that indeed backward selection has been used. One can do either all subsets or stepwise regression (such as backward or forward selection), but this describes two different approaches. Mundry 2011 and, more recently, Harrison et al. 2018 have reviewed different model comparison techniques and came to the same conclusion that any stepwise selection should be (and has been in the literature) “heavily critisised“ (Mundry 2011) and “ranking via AIC of all competing models is preferred” (Harrison et al. 2018). Therefore, I would strongly recommend the authors to redo the model selection part using all subset regressions (a personal hint: the R-package MuMIn has a function called “dredge” which should do a good model selection using all possible model combinations; different ICs are possible).
Line 250: with 34 observations only, the AIC value is not giving appropriate values and the use of AICc should be preferred.
Line 253: the R² value is a model fit value which can be calculated for LMs. However, as Harrison et al. 2018 summarised, even a high R² value does not necessarily tell us anything about the validity of the model. Further model fit measures are recommended (e.g., does the global model meet all model assumptions for an LM? Did the authors inspect residuals?). Another idea for the proportion of variance explained by each variable could be hierarchical partitioning (Lee and Nelder 1996, see package hier.part in R).
Lines 255-258: Multicollinearity should be assessed with the global model, not with the best model. Existing correlations already affect model selection not just the results and parameter estimates (I am citing Harrison et al. 2018 again, as this is a review and will lead you to further literature if needed). One can either test it with correlation tests (see my comments above to lines 245-246 for which results are missing) or with VIFs (but with all parameters, not just the ones in the best model). It is not necessary to do both.

Results:
The results are well presented. A figure showing the direct relation between AA and the best indices would improve the presentation of the results further. Model reporting including estimates is missing, preferably from the global model (Harrison et al. 2018). So far, readers get no idea on the direction of the effects.
Specific comments:
Line 272 and Fig. 1 suggest that 1989 has the lowest AA value. I do not understand this statement as both Fig. 2 and Table S1 suggest that the values of 2000 and 2012 are even lower (-5.28 compared to -6.02 and -5.94, respectively). Please clarify.
Line 276: please consider again that the analyses are not all-subset regressions
Line 279: multicollinearity should be checked before model selection and thus can be reported in the methods or in the beginning of the results
Line 281: It seems that the correlation coefficients have been reported (sorry for stating above that I did not find them), but again for the best model and not for the global model. This is confusing as I understood from the methods that correlation tests were done before model comparisons which is also why I was missing the results. Correlation tests of the final (“best”) model are meaningless as a correlation between two variables would violate model assumptions. Please note that a test for correlations should not only include the indices among each other, but also each index against year. Last, could the authors please explain why they retain the “year” in their model set (in the method section)?

Discussion:
The discussion is very comprehensive and discusses the different times and locations of the indices. The authors put a lot of effort in comparisons and explanations which I highly acknowledge. Some parts may even be a bit long, but I personally do not mind this too much. Some parts could also profit from slight language improvements.
Specific comments:
Lines 313-314: citation missing from other European studies mentioned
Line 376: I assume that the authors mean “a high abundance of insects”?
Line 494: this is the first time Fig. S3 is mentioned, while it would be worth mentioning every time the discussion or results describe one of these indices. I would suggest that this figure should best be included in the main text (results section) and is more important for readers than the current Fig. 3 which can be placed in the appendix instead.
Line 539: last letter, “t” – “the”?
Line 541: please insert “be” between would and easy
Line 543: “and to update annually” – it is not clear to me whether the AA index, the baseline or the database itself should be updated

Literature cited:
Grimm-Seyfarth, A. et al. 2015. Earlier breeding, lower success: does the spatial scale of climatic conditions matter in a migratory passerine bird? Ecol. Evol. 5: 5722–5734.
Harrison, X. et al. 2018. A brief introduction to mixed effects modelling and multi-model inference in ecology. PeerJ 6:e4794.
Lee, Y., and J. A. Nelder. 1996. Hierarchical generalized linear models. J. R. Stat. Soc. Series B (Methodol.) 58:619–678.
Møller, A. P. 2002. North Atlantic Oscillation (NAO) effects of climate on the relative importance of first and second clutches in a migratory passerine bird. J. Anim. Ecol. 71: 201–210.
Stephens, P. A., S. W. Buskirk, and C. Mart ınez del Rio. 2007. Inference in ecology and evolution. Trends Ecol. Evol. 22:192–197.

Reviewer 2 ·

Basic reporting

The article is clear, though the discussion is a bit tiresome and not very well organized (also within subsections).

Figures and tables lack headings and are not understandable without refering to manusctipt.

Bird raw data (subject of research) are not provided, but only some final statistics.

Experimental design

Problem of sample size which makes estimation of the lower part of the cumulative arrival curve unreliable. There were 21 years (out of 34) which had no more than 100 birds. This makes less than 5 birds (in some cases only 1 bird) for estimation of the lowest 5% of arrivals.
I suggest to start the curve at a higher point, preferably at the lower quartile (25%).

The Adjusted-R squared value, rather than the overall R-squared, should be used when presenting the results of a multivariate regression model.

Validity of the findings

Conclusions regarding the effect of climate variables are mostly speculations. This should be stated and conclusions shoule be presented in a more cautious manner.

Additional comments

Nice innovative approach for estimating spring arrival patterns.

Annotated reviews are not available for download in order to protect the identity of reviewers who chose to remain anonymous.

Reviewer 3 ·

Basic reporting

No comment

Experimental design

No Comment

Validity of the findings

No Comment

Additional comments

The authors have used a 36 years data set of captured willow warblers at a banding station in northern Poland to investigate the annual variation of spring arrival dates in relation to a dozen of climatic variables potentially reflecting the conditions of the species breeding, migratory and wintering grounds. The manuscript is clearly presented and the analyses are straightforward. In addition to the commonly used ways to estimate arrival dates (percentiles of e.g. 5%, 50%, ect) they use a rather novel metric, Annual Anomaly (AA), based on the cumulative deviation from the mean distribution of the whole study period. As in many previous studies of willow warblers conducted over times periods overlapping the present study, the authors find that the spring arrival dates have advanced by 5.4 days. A substantial fraction (59%) of the inter-annual variation in arrival dates could be explained by a set of the climatic variables with fairly similar contribution (Table 1). Interestingly, these included conditions from throughout the annual cycle, including the climatic condition during the previous breeding season, autumn migration, winter and spring migration. That these quite general variables could explain such a high proportion of the annual variation in arrival date is quite remarkable. This might be due to the fact that the inter-annual variation in arrival data in this data set was quite substantial (late and early years occur throughout the study period) and that the explanatory variables themselves were mainly uncorrelated (Table 2) providing sufficient statistical power to disentangle the effect of the explanatory climatic variables. Overall, it is a nice study worth publishing.

Detailed comments.
My main question relates to the composition of willow warbler subspecies in the data set. The study site in Poland is located in the migratory divide between the subspecies trochilus (wintering in western Europe) and acredula (wintering in eastern and southern Africa), see e.g. Bensch et al 2009 (Molecular Ecology). The birds captured at the banding station is supposedly a mix of these populations. I would guess that their divergent migration routes and wintering areas will be quite differently affected by the analysed climatic variables, e.g. trochilus in West Africa should perhaps not be affected by the Indian Ocean Dipole. I understand that it will be impossible to disentangle these birds into their subspecies, but it would be good to get an idea of their relative proportions. For example, the banding recoveries from the study site (spring birds) could be used to say whether the data mainly consists of trochilus, acredula or both. That you may have a mix of willow warblers with different migration routes and wintering areas should be mentioned in the introduction.

Several studies have examined variation in annual spring arrival dates of willow warblers in Europe, during time-periods overlapping the present study. The observed advancement of arrival dates are quite similar across the studies. It would however been interesting to see how well this local data set of annual arrival dates from Poland correlates with the arrival dates obtained from other places in Europe. For example, if your birds are mainly acredula, I would expect stronger correlations with arrival dates from e.g. Finland than from the UK.

Lines 424-425. Check species names, Pied flycatcher is Ficedula hypoleuca.

Line 490. Consider to replace “outstandingly” with e.g. the earliest.

Lines 534-535. I agree that AA seems to be a good way to measure annual arrival dates. However it would be useful to see how it correlates to the other measures, and whether its variance is smaller (assuming the others have more measurement errors).

It was indeed unfortunate that you had to exclude 2011 due to low sample size since this year showed remarkable delays in many migratory species wintering in eastern Africa (Tottrup et al 2012, Science).

---

## Round 0.2 · Minor Revisions

· Academic Editor

Minor Revisions

Although one reviewer (r2) indicated "major revision", I regard the comments as minor, hence my recommended decision on this revised version. I look forward to receiving your comments and revisions soon.

Reviewer 1 ·

Basic reporting

The article meets the standard, uses clear and professional English, cites appropriate litarature and shows excellent figures and tables. Raw data are shared.

Experimental design

No comment. Method is now described very well.
Very minor comments see below which might be added.

Validity of the findings

No comment

Additional comments

This is an excellent revision and the authors took over most concerns and implemented most suggestions. I particularly appreciate the testing of the many suggestions I gave.
I will first answer to the five specific points:

(1)
I particularly appreciate the test with local precipitation. I agree that these are too many variables. However, I would at least add in the discussion that NAO and local precipitation were positively correlated at the study site during the respective time, as this makes interpretation complicated whether the most important factor is NAO or local rainfall. That should at least be mentioned, now that the authors made all the effort.

(2)
I hope that the authors did not get me wrong – I highly appreciate the selection of the indicators by ecological meaning and not just by taking anything that is available. This was never questioned and, again, I am totally fine with the parameter selection. My only question was the collinearity, because I was missing it in a place where it should come. I highly appreciate the adaptations by the authors. Also, I am entirely fine with some collinearity which is to be expected in time series and ecological data, I was just wondering about the strength. The way it is presented now strongly improved the manuscript.

(3)
I also highly appreciate the test for linear versus quadratic terms and agree with the authors’ decisions. Only for one term, deltaAIC would be larger than 2, and I agree that this seems complicated to explain (through not impossible). However, as I believe that more people could question that, I would suggest to add a statement to the text that this has been tested and it did not lead to better AIC (I leave it to the authors how to describe the one exception – though it is only 2.49, so to be discussed whether the limit of 2 should be seen too strictly).

(4)
All subset regression: Great job! I highly appreciate it!

(5) I agree with the authors.

Further short comments:
Regarding the alternative approach I suggested: It is highly likely that this does not work for backward selection. Backward selection is highly sensitive to the order of the variables. If you change them (automatically through this approach), the result must differ (almost for sure) from whole model selections. If the authors decided to not touch precipitation issues, I am fine with the original approach.
Weather station: Great! Good luck with the local weather station.
I also highly appreciate the use of AICc, even if it shows identical results. For such a rather small subset, AICc just provides a better value (and sometime it indeed changes the order of the best models).
Figures and Tables are excellent.

Reviewer 2 ·

Basic reporting

No comment

Experimental design

No comment

Validity of the findings

No comment

Additional comments

Most comments were referred to and corrected and the statistical analysis has been modified and rerun. I find this manuscript innovative and important.

Some minor comments:

1. LL 386-388. You mention a change of 0.27 days a year of the first 5% of arrivals and of 0.17% of the first 50%. That is, a 60% difference between the two precentiles. Although this might indicate a stronger effect of climate change on the lower edge/ more “adaptive”/ ”non-conformistic”birds, this may be an indication of the unreliability of measures based on a too small a sample for the estimation of the 5 prectile. Please, point that out. Please notice that this posibility is supported by the much smaller change for the 10th percentile (Table S2).
Additionally, I suspect that you got some mistake in this table, as the overall change of -7.9 days is identical for the 10th precentile and the median, but the slopes of these two precentiles are different: -0.17 versus -0.22. How come?

2. You exapnd quite a bit about checking for multicollinearity among explanatory variables. This is somewhat redundant because multicollinearity, if exists, causes inflation of the variance which may obscure possible relationships between the response and the explanatory variables. Therefore, any statistically significant results obtained, would be in fact conservative in case of multicollinearity. Eliminating a potential problem is, of course, desirable in order to receive a more complete picture of the statistical relationships. However, the elimination of multicollinearity could only enhance relationships already revealed in a model, not dismiss them.

3. Figures headings are long and difficult to follow. Figures in particular should be self-explanatory as words are in the manuscript’s text. Please add legends with images instead of the many words (e.g. ”green arrows”, “thick lines”, etc. should be represented by an image).

Figure 1: Please simplify subheading (A) and divide the long sentence into 2-3 short ones. Additionally, please add a legend explaining the different lines/arrows, rather than writing it in the heading’s text.

Figure 2: It seems that numbers and words of headings got mixed up in your PDF file. Please correct headings to:
(A) Spring migration in 1997 in relation to the regression line of overall spring migration during 1982–2017;
(B) Spring migration in 1989 in relation to the regression line of overall spring migration during 1982–2017;
Please add a legend presenting the different lines/colors and shorten the worded explanation.
Please break the last sentence into two instead of using “which” (start with a picture of “light grey”, explain what it represents, and then turn to the dark grey.

Figure 3: Please add a feature legend with images (a line with circles, with triangles, etc.).

Figure 4: Please add the p-value to each of the regression’s graphs.

Tables 1 & 2 belong in the supplementary material. They are not necessary for the coherency of the manuscript.

Table 3: Please limit wording. Details regarding the statistical outcomes, such as F-value, Adj -R2, etc. are self-explanatory and the extra-wording is redundant. i.e., it would be enough to write:

Best model statistics:
Overall F =…, p-value =…, Adj-R2 = …

---

## Round 0.3 · accepted · Accept

· Academic Editor

Accept

Thank you for addressing the reviewers' comments so comprehensively.